# Kinesthesia and Temporal Experience: On the ‘Knitting and Unknitting’ Process of Bodily Subjectivity in Schizophrenia

**DOI:** 10.3390/diagnostics12112720

**Published:** 2022-11-07

**Authors:** Camilo Sánchez, Marcin Moskalewicz

**Affiliations:** 1Philosophy of Mental Health Unit, Department of Social Sciences and the Humanities, Poznan University of Medical Sciences, 61-701 Poznan, Poland; 2Institute of Philosophy, Marie Sklodowska-Curie University, 20-400 Lublin, Poland

**Keywords:** kinesthesia, lived movement, temporal experience, lived time, phenomenology, functional connectivity

## Abstract

This paper proposes a phenomenological hypothesis that psychosis entails a disturbance of the two-fold process of the indication function of kinesthesia and the presentification function of touch that affects the constitution of bodily subjectivity. Recent functional connectivity studies showed that the increased synchrony between the right anterior insula and the default mode network are associated with psychosis. This association is proposed to be correlated with the disrupted dynamics between the pre-reflective and reflective temporal experience in psychotic patients. The paper first examines the dynamic nature of kinesthesia and the influence touch and vision exert on it, and then the reciprocal influence with temporal experience focusing on the body’s cyclic sense of temporality and its impact on physiology and phenomenology. Affectivity and self-affection are considered in their basic bodily expressions mainly through the concepts of responsivity and receptivity. The overall constitutive processes referred to throughout the article are proposed as a roadmap to develop body-based therapeutic work.

## 1. Introduction

The main purpose of this article is to articulate an approach to schizophrenia based upon key phenomenological findings concerning the constitutive function of kinesthesia and the synthetic processes resulting in temporal experience and to contextualize them with relevant neuroscientific findings on functional connectivity. The first part of the article characterizes the concept of kinesthesia, the second part presents some experimental approaches to kinesthesia, the third part elaborates on phenomenologically relevant aspects of cyclic temporal dynamics in living organisms, the fourth part speaks of affectivity as an essential element integrated into movement, the fifth part synthetically presents schizophrenia as a disorder of temporal experience and associated functional connectivity impairments, and the sixth and final part describes the disturbed dynamics between kinesthesia and touch in a psychopathological context as the core pathology of schizophrenia.

## 2. Kinesthesia and Animation

The concept of *kinesthesis* was introduced by Bastian [1] referring to the sense of movement, contributed by the receptors in the skin, muscles and deep limb structures, including joints, fasciae, and tendons. The term kinesthesia or kinesthesis for Proske and Gandevia [2] entails the sense of position. Although they are phenomenologically distinct, the sense of movement entails the sense of position, i.e., part of the proprioceptive content of experience.

Phenomenologically, kinesthesia consists of the actual felt sense of movement, a tacit but main constituent of the lived body or Leib. It is accessible to the agent and cannot be inhibited ([3], p. 56). Thus, if one attends to the experience of movement, it implies a qualitative dynamic that involves temporal, affective, and spatial aspects, among others. When observed as a three-dimensional phenomenon, it also involves a kinetic aspect. The spatial characteristics of embodied movement thus has a double meaning—it can be kinesthetically felt or kinetically perceived. The distinction is based on the fact that third-person phenomena are perceived as objects, whereas first-person phenomena are felt by the agent. However, an agent can kinesthetically perceive one’s movement’s quantitative dynamics and can also kinesthetically feel its qualitative dynamics ([4], p. 38).

The standard scientific approach characterizes movement kinetically by quantifiable concepts, while the kinesthetic qualitative characterization of movement is non-linguistic. Sheets-Johnstone highlights ([5], p. 7) that consideration of movement requires a terminology that apprehends its characteristic continuous flux and change in contrast to sensations amenable to point-to-point consideration. As she put it: “*In the course of everyday life—walking down the street, sitting in a seat… we do not experience movement as a discrete series of moment-to-moment, now-here, now-here, kinetic events (…) we do not have temporally punctiliar, spatially punctiliar kinesthetic sensations. Briefly, what we have are kinesthetic sensations of a qualitative kinesthetic dynamic*” ([6], p. 367).

Brian O’Shaughnessy’s *Proprioception and the body image* [7] illustrates this ‘pointillist’ conception of postural sensations conceived as a moment-to-moment postural change. O’Shaughnessy analyses proprioceptive content as the foundation of embodied spatialization through the short-term body image (STBI or ‘I’), and proprioception, as opposed to kinesthesia, is suited to such pointillist analysis. The pointillist conception is not only inadequate to account for the cognition of animate beings (*Animation* ([5], p. 10) alludes to the fundamental kinetic realities that link cognition and affectivity, and enable synergies of meaningful movement.) but makes kinesthesia and proprioception equivalent, and although they share sensory content, they serve distinct functions. Following the pathway of evolutionary biology considered by Sheets-Johnstone [8,9], proprioception was a surface recognition sensitivity that complementarily served movement through deformations and decompressions, but then evolved from outer sensory organs (such as cilia and slit sensilla) into organs sensitive to inner-body stress (such as chordotonal organs) [10,11,12,13]. This transformation gave way to a direct awareness of movement, i.e., kinesthesia. In the 19th century, this faculty was called ’the muscle sense’ [1,14].

Maxine Sheets-Johnstone [4,6] rejects the pointillist conception based on the qualitative kinetic–kinesthetic–tactile dynamics as the central feature of embodied movement, attributed by Husserl in his *Cartesian Meditations* [15]. The latter grounds the distinction between sensations and dynamics: Sensations have no place in a qualitative dynamic as they are antithetical to the experience of movement, while the qualitative dynamic flows forward in a continuum, in which the unfolding of embodied movement constitutes its temporality.

Following Darwin, Sheets-Johnstone ([5], pp. 8–9) states that the morphological structure of living organisms is one of the basic assets for the individuation of their environment. It conditions the possibilities and hierarchy of corporeal movement, and the experience of movement structures a sense of the body [16]. Both conditions are factors for the individuation of the world and, complementarily, of the constitution of an animate and living body. This constitution process is mainly attained in terms of movement [17,18,19], as the efficiency of the latter is an evolutionary determinant for survival ([20], p. 4) through meaningful synergies [4,21,22,23].

Individuals as living, active organisms are kinesthetically receptive to bodily tensions, postural attitudes, and, broadly, to the qualitative kinetic dynamics, which unfold continuously and belong to recessive awareness. Sheets-Johnstone exemplifies the agent’s familiarity with their own dynamics through brushing one’s teeth: If an agent does it by himself, its dynamics unfold probably unnoticed; if it was done by somebody else, it would feel strange and intrusive. The familiarity of one’s own experience is structured upon the attenuation effect [24] because when one brushes one’s teeth, the actual sensory feedback is attenuated as it is rhythmically and thus temporarily coordinated with the efferent copy of the brushing movement. The sensory feedback is characterized by temporal and spatial factors such as rhythm, cadence, frequency, amplitude, and range ([5], p. 12), on which familiarity or strangeness is also anchored.

“*Familiar dynamics—tying a knot, brushing one’s teeth… writing one’s name…—are interwoven in our bodies and unfolded along the lines of our bodies. They are kinesthetic/kinetic melodies in both neurological and experiential senses (Luria 1966, 1973). When we pay attention to these familiar dynamics, to our own coordination dynamics (Kelso 1995; Kelso and Engstrom 2006), we recognize kinesthetic melodies; they bear the stamp of our own qualitatively felt movement patterns, our own familiar meaningful movement synergies (Sheets-Johnstone 2009 a, b)*”([4], p. 24)

As recognized by Russian psychologist Luria ([25], p. 176), kinesthesia unfolds in qualitative patterns that, when reinforced, structure a distinct individuality; these patterns articulate throughout the whole living organism, constituting bodily ‘avenues’ of movement. A network of articulated patterns of qualitative dynamics constitutes a personal sense of body through kinesthetic memory (‘Body memory’, [26]). According to Sheets-Johnstone ([13], chap. 5), movement developmentally precedes the constitution of individuality (‘self’), and the constitution of the latter is accomplished when the infant learns to move effectively through the environment ([20], p. 24). Therefore, the morphological structure, kinesthesia, and the complex of individual qualitative dynamics support a sense of body and movement possibilities for the particular living organism.

## 3. Experimental Approaches to Kinesthesia

It is well-established that kinesthesia is influenced by vision [27,28,29,30,31,32] and touch [33,34,35,36]. The experimental approach to kinesthesia has frequently focused on vibration-induced illusions [37] by artificially producing different frequencies of tendon vibration [38], e.g., on two-arm matching tasks. To account for these kinesthetic illusions (movement or flexion), Proske and Gandevia ([20], p. 1158) proposed they originate by activating the muscle spindle primary endings. Based on the latter hypothetical assumption, brain-imaging research has been looking to find the specific activity associated with the induced illusions.

It was found that the primary muscle afferents project into the primary sensorimotor area ([39], p. 75), also in animal models; some of these afferent projections concern the supplementary and cingulate motor areas ([39], p. 75). The muscle spindle inputs Brodmann area 3a, which connects to the primary motor cortex, supplementary and cingulate motor area; and the ventrolateral/ventral posterolateral thalamic nuclei also project to the cingulate motor area, the premotor and primary motor cortex. Muscle stretch or passive joint rotation can evoke activity in the cingulate and supplementary motor area, and the premotor and motor cortex. Most authors point to the parietal and motor cortex activity as the main associations of kinesthesia induced by either active or passive movement.

From the experimental perspective, kinesthesia entails a body structure articulating the experienced content throughout the body, to which the recent brain imaging techniques can contribute [40,41,42]. Following Longo and Haggard [43], there are mainly three neurobiological associations to the required body structure: (i) the somatosensory map or S1 (The left parietal cortex has been associated with the image of the body [44]) [45], (ii) the body musculature or M1, (iii) the postural schema [46]. Proske and Gandevia contrast the latter body model with the conscious body image. The former was experimentally tested by asking subjects to judge the shape of a body part and then match it with a visual template [47]; as most of the subjects judged accurately, the supposed impenetrability of the body model was contrasted with the malleability of the body image, to which kinesthetic dynamics contribute significantly [43].

The neurobiological and brain image research associated with a body structure, or ‘representation of the body’ was reviewed lately by Naito and colleagues [48]. Frequently the experiments entail vibration applied to the wrist muscles of one arm while the subject is lying supine in the MRI scanner. The movement illusion produced by the vibrations is reported by the subject and can be quantified. If the illusion occurs, the activity in the motor and the somatosensory area is recorded. This activity comprises the primary motor area (M1), dorsal premotor cortex, supplementary motor area (SMA), cingulate motor area, basal ganglia’s sensorimotor and cerebellum’s motor area. Some authors [49] pointed out that the activity in M1 during the wrist flexion illusion was less than half of the one during actual wrist flexion, and it was displayed on the somatotopic hand region of the motor cortex. It was concluded that M1 is part of the associated network of kinesthetic experience and, bearing in mind its relevancy in voluntary movement, also a tight link between perception and action.

When the vibration is applied to wrist flexors while holding a ball, the subject has the illusory experience of moving the hand with the ball [44]. The brain activity resulting from the illusion involves the left hemisphere’s inferior frontal and parietal regions. The general parietal function seems to integrate the content of movement and touch, while the posterior parietal is related to the influence of vision on kinesthesia. Finally, it has been hypothesized that the inferior branch of the superior longitudinal fasciculus is correlated with the constitution of body ownership, including foreign objects, e.g., rubber–hand illusion, as well as body and self-awareness.

## 4. Cyclic Temporal Dynamics of Living Organisms

Every awareness entails a temporal sense that is constituted by specific retentional–protentional dynamics. A fundamental phenomenological finding is that three components constitute temporality: protention which corresponds to anticipation of what is coming, presentation which corresponds to the ever-reaching ‘now’, and retention, which corresponds to the remains of what has just happened ([50], p. 93). Bearing in mind the animated nature of human awareness, i.e., as it couples affect and cognition through kinetic realities, animation functions as a standard matrix for the felt sense of temporality and movement, as Sheets-Johnstone also underscores.

“*When we listen to our ‘inner awareness of movement‘, we find a distinctly felt temporal flow or present transmission that is constituted in the very process of being created.*”([4], p. 34)

In other words, the sense of temporality unfolds along with the kinesthetic flow of movement, and the kinesthetic flow unfolds temporality. The American composer Roger Sessions ([51], p. 108) informally conveys the entanglement of the experience of bodily movement and time in the following words: “*It seems to me that the essential medium of music, the basis of its expressive powers and the element which gives it its unique quality among the arts, is time, made living for us through its expressive essence, movement. Time becomes real to us primarily through movement, which I have called its expressive essence; and it is easy to trace our primary musical responses to the most primitive movement of our being—to those movements which are indeed at the very basis of animate existence. The feeling for tempo, so often derived from the dance, has in reality a much more primitive basis in the involuntary movements of the nervous system and the body in the beating of the heart, and more consciously in breathing, later in walking*” ([51], p. 105). Hence, awareness is scaffolded on kinesthesia and body qualitative dynamics which entail a rhythm and pattern of protention and retention. In agreement with Gallagher’s proposal [52], a distinction between action and movement is proposed i.e., movement corresponds to a pre-intentional level of constitution of meaning and is framed in ‘finer’ or ‘elemental/integration’ time, while action corresponds to an intentional level of constitution and is framed in broader ‘narrative’ time.

Following Husserl’s phenomenology [50], the implicit experience of time is pre-reflective and explicit is reflective. When someone is casually scrolling through the posts on social media their temporal experience is implicit, but when s/he monitors the time passed while scrolling, it becomes explicit. Through the pre-reflective unfolding of experience or auto-affection, minimal subjectivity or mine-ness sphere is constituted ([53], p. 168). The prerequisites of the implicit unfolding of temporal experience are passive synthesis [50,54] and conation [55]. The notion of passive synthesis refers to the fact that phenomenological impressions are organized passively by the mind and affect the subject before they are consciously noticed. The notion of conation integrates spontaneity, affectivity, and attention, and entails the intentional structure of cognition and behavior. This intentional structure refers to the directedness of the mind towards objects, which is a basic attribute of all mental states [56]. Both, passive synthesis and conation account for the continuity and variability of experience, correspondingly.

The reciprocal interdependence between kinesthetic and temporal dynamics is exemplified by the rhythm of the processes in the living organism, i.e., heartbeat, breath, metabolism, etc., and vice versa [57,58].

All these homeostatic regulatory processes are integrated into the living organism dynamics of deficiency-desire-satisfaction instantiated by, e.g., the digestive and circadian cycle and the trophotropic–ergotropic phases. These cycles that sustain the living organism’s dynamics are finely attuned with the environment, e.g., hormone secretion and energy availability according to the circadian rhythm, in which the hypothalamus is greatly involved ([58], p. 49); the nucleus suprachiasmatic is associated with rhythmic movement, circadian cycle, and food/liquid intake while the nucleus ventromedialis is associated with insulin, glucose, temperature and energy levels.

This broad primitive cyclic dynamic scaffolds the two-sided axis of temporality and movement experience. The physiological cycles set a sense of rhythm affecting the stream of consciousness, e.g., breath and heartbeat, from which the temporal experience is constituted ([50], p. 93). Through the entrainment of the stream of consciousness, the cyclic temporality sets a pace on the felt sense of time and movement. It has been hypothesized [59,60,61] that the sense of duration results from the integration of rhythmic bodily signals mediated by the parietal and insular cortex, operculum, and middle temporal gyrus. One of these founding bodily signals is interoception i.e., a sense of the physiological condition of the body which integrates bodily signals in certain brain regions, promoting homeostasis and awareness, which contributes to pre-reflective self-awareness. Part of the latter hypothesis highlights interoception as main contributor to the temporal continuity of embodied self ([58], pp. 50–51). In this article the cyclic physiological and deficiency–satisfaction bodily dynamics, are laxly considered as contributors to the sense of cyclic temporal experience; nonetheless, future research has to address if the latter are part of the stream of consciousness.

These cyclic temporalities respond to different periods. The latter physiological processes instantiate the longer-range dynamic of deficiency–desire–satisfaction ([58], p. 57) and contribute to the even longer-range temporality of body memory [26,57,58,62]. The movement-based and bodily structured habits [23], unfold periodically according to environmental cues. They often unfold in all-encompassing patterns: on the one hand, they guide the whole organism’s dynamic according to specific cues, and on the other, they evoke dynamical patterns that have been lived years or even decades ago.

The second condition entailed by the pre-reflective temporal experience is conation. The notion of conation integrates spontaneity, affectivity, and attention, which entails the intentional structure of cognition and behavior. Conation refers to the wide behavioral spectrum encompassed by the drive/urge to move and take action. It is a factor in the constitution of the intentional arc (This term proposed by Merleau-Ponty [63] highlights the individual’s directional structure in perception and movement patterns e.g., motor schemas or gestalten, which through body memory display operative responses), which endows with continuity and affective consistency ([63], p. 140). Conation contributes to sustaining and advancing the dynamic of deficiency–desire–satisfaction and regulates the physiological cycles. The teleological structure of action serves as a template to promote and advance the cyclical dynamic because, by opening a ‘time-span’ and an ’appetitive tension’, the need and strive prompt protention towards fulfillment ([57,64], p. 14). Complementarily, the qualitative dynamics enhance pre-reflective self-awareness and may also function as a prompter of renewed protention. The latter highlights that the pre-reflective bodily temporality entails a sense of rhythm. Only when an individual or group reflexively coordinate time, the temporal experience becomes explicitly reflective and linearly ordered. Pathological behaviors entail a desynchronization between the two, such as conative slowing down in depression ([58], pp. 61–62) or speeding up in mania ([58], p. 61) or their total disintegration [65,66].

## 5. Affectivity and Self-Affection

Affect is another key element for the constitution of subjectivity and a frequent component of psychopathological manifestations. Husserl’s phenomenological tradition categorizes affect as a passive constituent of experience mainly associated with retention, but it also functions as a protentional force constitutive of the pre-reflective realm. Following Husserl, ‘a positive affective force is the fundamental condition of all life,’ and along with it, he characterizes bodily receptivity ([67], pp. 76–79) as one of the lowest activity stratums of subjectivity, constituted by affect and movement ([67], pp. 76–79), where turning toward is one of the basic affective responses.

In evolutionary biology, a notion parallel to receptivity is responsivity. It accounts for an almost universally shared feature of life, i.e., the capacity to be affected by the environment and respond to it ([68], p. 28). From plants through bacteria to animals, bodies move toward or away from others or the environment, e.g., notochord and vertebrates have constituted these basic affective responses through kinesthetic unfolding ([9], p. 8), and this organic affectivity amounts to elemental kinetic unfolding giving way to affective proto-dialogues flowing in patterns of movement.

By participating in the kinetic inter-corporeal sense-making with all living organisms or “*kinetic bodily logos*” ([13], p. 444) and moving toward what is self-affirming and away from what is not ([8], p. 46), these kinetic–tactile–kinesthetic–affective patterns are opportunities for pre-reflective self-awareness and other awareness through movement ([8], p. 46).

Affect guides organisms by modulating the cyclical-physiological processes, advancing the dynamic of deficiency–desire–satisfaction, and, along with the latter, the qualitative dynamics of the whole body. According to Depraz and Varela ([69], p. 69), it mainly appears through movement and disposition. Movement instantiates the affective force through two poles of animation: ‘towards’ and ‘away’. Disposition is part of the continuum of movement and bodily specialization, and it is defined on the basis of facial gestures and posture complemented by the autonomic response.

The latter proposal highlights affective valence as the common currency in animated life. The polar orientation of qualitative dynamics constituted by valence, in turn, co-constitutes self-affection. Thus, the qualitative bodily dynamics co-constitutes affect, temporal, spatial, and movement experiences. This dynamic instantiated in temporal patterns of movement contributes to how a living organism responds to its own aliveness in the environment affecting itself. Depraz and Varela ([8], p. 74) propose that the affective polar dynamic contributing to self-affection is continuously embodied as an availability for action, i.e., posture. Similar to O’Shaughnessy [7] and Sheets-Johnstone [4,5,6], Depraz and Varela [8] extend their conceptual/phenomenological findings to the phylogenetic scale, alluding not only to sensory–motor invariants but also to valence sedimented in brain connectivity ([8], p. 74). It is essential to review the cortico-cerebellum–basal ganglia connectivity by evaluating, identifying and phenomenologically characterizing the phylogenetic and ontogenetic valences affected by this connectivity.

The rhythmic patterns of temporal experience and affect in movement, which contribute to the bodily constitution of subjectivity, are naturally available to bodily resonance with rhythms in music, physiological cycles, and other moving bodies with which one can share time through inter-corporeal synchronization.

## 6. Schizophrenia as Disturbance of the Temporal Experience and the Triple Network

Many contributions to phenomenological psychopathology focused on temporal experience as necessary to understand, research, and study mental illness and psychosis [18,70,71,72]. Minkowski’s early contribution to this theme already focused on the tension between the reflective linear conceptualization of time and its intuitively ‘irrational’ experience [70].

Minkowski’s central thesis on mental illness was that it stems from a dissociation between the linear clock-time and the irrational individual experience of time, which manifests through the inability to conceive an open future within the linear (‘objective’) framework of time. Similar conceptions of mental illness were proposed by psychologists [73,74], although focusing rather on the present experience.

Minkowski’s phenomenologically grounded insight was that a psychotic patient loses the pre-reflective sense of heading towards the future, a part of an implicit and social sense of becoming, and synchronicity with the environment and fellow others ([70], pp. 59–63). Based on the same sense of implicit heading forward, Japanese psychiatrist Bin Kimura later distinguished three psychopathological orientations towards the future: post-, intra-, and ante-festum. It means after, during, and before the ‘*feast*’, referring to the patients’ experienced sense of a decisive event that took, is taking, or is about to take place in their life [18]. The ante-festum associated with psychotic experience consists of protentional instability regarding the whole self and is manifested through ontological anxiety, which is similar to Conrad’s description of Trema regarding the prodromal phase in schizophrenia.

Minkowski defined schizophrenia as a ‘loss of vital contact with reality’ marked by the feelings of immobility and ‘morbid rationalism’—the latter also referring to an obsessive reflection on Weltzeit or mathematical clock-time as a trade-off for the loss of Ichzeit or fluid becoming, often accompanied by disbelief in time. Following Fischer [75], this orientation frequently manifested by psychotic patients was called sinnlos ([70], p. 186). It could manifest itself as a delusional belief that the clinic staff manipulate time, due to the patient’s loss of the pre-reflexive experience of time remaining only with a reflexive one, hypertrophied.

“*I love immutable objects, things which are always there, and which never change… The past is the precipice. The future is the mountain.*”([70], p. 279)

The latter extract of a patient illustrates the paradoxical characteristic of psychotic experience, i.e., on the one hand, immobility, but on the other chaotic flux of change; this characteristic grounds part of the suffering lived by patients, genuinely expressing a deep longing for individual stability and cohesion through their appreciation of the permanence of environmental objects. It is constant in psychotic patients to point to the immobility and lack of flow in themselves and of time ([70], pp. 285–286) ([75], p. 545). These mentions testify to a temporally unstructured experience, as the passive synthesis constituting this experience has been unwoven by the pathological process, leaving the patient with a past–present–future structure unavailable, a deeply impaired awareness as there is no sense of flow or unfolding of the pre-reflexive experience, and therefore, bodily qualitative dynamic also impaired in their free flow structuring movement.

“*Figuratively speaking it seems years since I was out in the normal world… I never know any moment what is going to happen. It’s the most terrible outlook I’ve ever had to look to. It’s all perpetual. I’ve got to suffer perpetually.*”([76], pp. 617–618)

This report verifies the loss of the temporal structure of experience and its retention–protention dynamics necessary for the intentional function (operative intentionality and intentional arc in Fuchs [77]), which serves as a platform for human dealings with the world and others. The latter, in broad terms, is part of the reason why there is a loss of contact with reality and appalling suffering from the patients.

The loss of the temporal structure of experience supports Minkowski’s thesis and Schilder’s similar claim ([78], p. 265) that psychotic patients are unable to become in time due to their lack of orientation toward the future since becoming can only take place through a healthy flowing retentional–protentional dynamics and pre-reflexive realm which constitutes basic individuality or minimal self. Following Minkowski, pathological desynchronization prevents the individual from sensing and synthesizing its pre-reflexive temporal experience (Ichzeit) and a reflexive one, socially shared (Weltzeit). Even though psychotic patients have shown ([79], p. 21) particular sensitivity to mis-attunement between their pre-reflective temporal flow and the reflective social one, they cannot synchronize the former with the latter, mainly dwelling in clock-time. A disturbance of inter-corporeal synchronization in schizophrenia is a likely source of the so-called praecox feeling, which psychiatrists still use for their diagnoses [80,81].

Recent research on phenomenological psychopathology [57,58,77,82] has further developed the thesis on schizophrenia as a pathology of the temporal constitution, and meticulously elaborated its consequences in terms of symptoms suffered by patients. On the other hand, recent experimental research on schizophrenia made significant advances in enlightening shared diagnostic, therapeutic, and conceptual issues. One of these is the functional connectivity [83] approach, which proposes to understand schizophrenia as an impairment of the functional integration of the brain.

One of the findings [84] was that a pathophysiological feature of schizophrenia consists of an impaired synchronization between the default mode (DMN) and the central executive networks (CEN) (DMN consists of the precuneus (PC), posterior cingulate cortex (PCC), medial prefrontal cortex (mPFC), and lateral parietal cortex. Its activity is associated with a focus on inner experience and inversely correlated to brain networks associated with external stimuli [85]. This network is deactivated with attention focused on a task [86]. And, CEN consists of the dorsolateral prefrontal cortex (DLPFC), posterior parietal cortex, medial frontal gyrus (MFG), superior frontal gyrus (SFG), ACC, paracingular gyrus, ventrolateral prefrontal cortex (VLPFC), and subcortical areas such as the thalamus. This network is involved in executive functions [87]). A disturbance in the interaction between three large-scale brain networks; DMN, CEN, and salience network (SAL) was found in psychotic patients [88]. Most recently, real-time correlation between item responses on the paranoid self-evaluation scale and activations in the DMN (precuneus and angular gyrus) in schizophrenia was described and contrasted with the activations during depression items processing and response [89] (SAL consists of the anterior insula (AI) and the dorsal anterior cingulum (dACC). It integrates sensory, emotional and cognitive projections for communication, social behavior, and self-awareness [90]; essential for regulating the rapid shift of focus)

Liu and colleagues [91] and other recent electroencephalography research have shown the importance of assessing non-linear relationships between frequency rhythms and distant brain centers. Dynamic functional connectivity-based analysis of schizophrenia [92] has been gaining importance, along with time-resolved fMRI research [62]. Looking to advance the early detection of psychosis and clarify the function played by the right anterior insula and some co-activating regions, Bolton and colleagues applied a resting-state fMRI on 25 subjects at risk of psychosis, according to basic symptom criteria [93], 18 subjects with attenuated or brief intermittent psychotic symptoms [94], ultra-high-risk subjects and 29 healthy controls. The co-activated patterns relevant to the dynamics between the triple network (DMN, SAL, CEN) were assessed in their temporal interaction.

It has been proposed [95] that a pathological interaction pattern of the triple network accounts for different psychopathological manifestations. Bolton and colleagues [88] evaluated the interaction of right anterior insula-driven networks for subjects in different at-risk-for psychosis conditions and showed that across intrinsic states the insular seed activates the whole resting-state scan along with other well-known networks. Nomi and colleagues [96] showed that CAP3 and CAP6 included visual and basal ganglia networks, and CAP6 may support dopaminergic paths implicated in insular activity [97]. Their results confirmed the regulating role of SAL on DMN and CEN activity, blending information [95,98] as proposed by the triple network hypothesis, which asserts a competitive interaction between these networks as if they were associated with individual dealings relative to the inner (DMN) and outer (CEN) realms [99]. Following these results, the insular seed evidenced a polar dynamic relative to the DMN, antagonistic in CAP1 in coactivation with SAL and temporoparietal network and agonistic (coactivation) in CAP2, which supports the results of Karahanoglu and Van De Ville [62] and also of Nomi and colleagues [96].

Ultra-high-risk (UHR) subjects showed longer CAP2 states, i.e., longer and greater activation patterns of right anterior insula and DMN, supporting earlier results of Satterthwaite and Baker [100] who also showed greater connectivity within DMN and diminished within SAL on young subjects with subthreshold psychosis spectrum symptoms. The DMN has been associated with mind wandering [101], self-awareness, source attribution [102], and self-guided and self-referential thought [103]. The right anterior insula has been associated with bottom-up processing through interoceptive paths, probably through the basal ganglia and via the CEN [104], supporting relevant regions coping with appropriate behaviors to environmental challenges. The functional connectivity approach led to interesting hypotheses, e.g., the proximal salience and source monitoring model [95,105,106], looking to account for various schizophrenia symptoms.

It is important to highlight the contrast Bolton and colleagues ([88], p. 7) showed between ultra-high-risk and healthy controls or subjects with basic symptoms regarding the balance interaction within the triple network. UHR subjects did not instantiate a competitive activity between CAP1 and CAP2, which could be associated with psychotic dissociation between the individual and social realms [107]. Bolton and colleagues [88] speculatively proposed that the loss of CAP1 regulatory activity might be associated with the weakening of CEN regulatory top-down activity and might give way to further disintegration of the triple network associated with the unfolding of self-disturbances in psychotic patients [108].

The anterior insula regulates the activity between DMN and CEN, which is disrupted before the onset of psychosis. Bolton and colleagues ([88], p. 7) evidenced that the UHR subjects had greater periods of synchrony between the right anterior insula and the DMN and lost the competitive dynamics between the synchronic activity of the SAL and DMN, and the CEN and insular activity (along with DMN deactivation). A similar dynamic pattern is evidenced in children, who have greater individual activity from each of the triple networks than synchronic activity between them [109]. This suggests a similarity between the triple network activity pattern in UHR subjects and children [100]. Wotruba and colleagues [110] suggest that the latter impairment in the triple network dynamics may be associated with the loss of contact with reality suffered by psychotic patients, along with a morbidly excessive awareness of the individual realm; the latter could be associated with the greater synchronic pattern between the right anterior insula and the DMN.

## 7. The Twofold Dynamics between Kinesthesia and Touch in Psychosis

A psychotic patient states: “*I feel I’m not continuous. I seem to be reborn every moment*” [75]. These words express an almost invariable feature of psychotic patients’ narratives—the loss of the unfolding dynamics of pre-reflexive temporal experience and, in consequence, their pre-reflective sense of self. No surprise, if there is not a constituted implicit temporal experience, there cannot be a constituted minimal self because it is through the former that the latter is constituted. Fischer concludes that ‘*psychotic states leave no trace of a past structure in the mind*’ ([75], p. 252), but the disturbance remains explanatorily intact even if the pathogenesis is traced to the original constitutional process or to a later pathological process. Either way, these processes entailed by the passive synthesis require study and experimental research, as they could enlighten the experimental findings towards a more accurate association pathway by truly integrating the core of the mental illness, i.e., the individual and collective suffering, so that the recovery of the well-being could be integrated into the goals of experimental research.

Husserl refers [111] to the product of the constitutional processes as sedimented layers on which the patient’s experiences ‘rest, but it seems that psychotic patients remain ‘plowing in the water’ regarding the latter, which is the reason why their embodied subjectivity requires rebuilding or recovery work. How are these layers of embodiment sedimented? Husserl’s findings refer to a twofold dynamic process between touch/haptic sense and kinesthesia ([111], pp. 146–154). The former serves the function of presentification while the latter the function of indication. Originally, they are blended into a structured embodied sense through passive synthesis. Through this knitting-like dynamics between sensing and being sensed, the constitution of embodied subjectivity can be achieved.

Psychosis not only erodes the layers of implicit Ichzeit but the ability to synchronize with others through interaction [57]. When an individual learns an ability through movement, they not only learn to move, but also has a felt sense of the open possibilities offered through kinesthesia (indication), and access its movement through sensory feedback, mainly tactile or visual (presentification). Thus, through the constitutive twofold displayed in movement, one sets one’s own pace or bodily rhythm, upon which one’s embodied subjectivity is structured, and the social synchrony of sharing time and presence takes place. Whereby, if the constitutive twofold process is disrupted or its product corrupted, much of what bodily subjectivity enables becomes disabled. Moreover, one may track down particular cases of mental disturbance into a continuum of the constitution of bodily subjectivity: having a pole of an absence of constitution (severe psychosis) and another of affected bodily subjectivity (milder cases). While the intervals around the former are due to severe cases, the ones around the latter could exemplify everyday life mental health issues, resulting from individual limitations (referring to aspects of individuality which escape willful government e.g., organic instinct-based reactions, this relation with alterity or otherness contributes to the constitution of selfhood ([112], p. 338)) and interpersonal conflicts. The proposed continuum is oriented toward developmental constitutional processes of embodied subjectivity grounded in the twofold dynamics, and along with it a group of phenomenological variables may serve as an experience-structuring axis. These latter phenomenological concepts, such as temporal experience (implicit, explicit), spatial experience (phenomenal field, life space), bodily movement (self-affection, hetero-affection), affect (pre-predicative/predicative modalizations) and intentionality (pre-predicative/predicative modalizations) should be integrated with the rest of the findings and associations from the genetic, molecular, cellular, physiological, and behavioral level. Thereby, the different psychopathological conditions profiled along the continuum are understood as individual outcomes of intersected variables from the proposed mental health assessment matrix. The presented matrix may serve as a conceptual framework orienting diagnosis and clinical care, it may also function as a theoretical background to support the classification issues, and finally, it may ground the development of therapeutical strategies.

## 8. Conclusions

This article proposed the twofold dynamic process between kinesthesia and touch as the ground from which psychotic disturbances emerge and, therefore, from which the therapeutic work has to be developed. The two main phenomenological axes, movement and temporal experience, are in deep constitutive connection with the twofold dynamics; the broad constitutive processes referred to throughout this work are proposed as a roadmap to develop the therapeutic body work to aid more traditional or technologically based pharmacological approaches [113], looking to rebuild and/or recover the unhealthy suffering embodied subjectivity.

Additionally, there is potentially two-way productivity between functional connectivity and the phenomenological psychopathological approach to schizophrenia, which has to be fully scientifically exploited. Framed by the triple network hypothesis of schizophrenia and enriched by the phenomenological psychopathology insight, some potential associations can be suggested. The default mode network (DMN) function is associated with unfolding of temporal experience based on the activity pattern in UHR subjects. According to recent neuroscience work on the DMN [114], the dynamic patterns of this network’s activity are correlated with temporal experience. DMN activity has been associated [115] with ‘mind-wandering’ (MW) conceived as an embodied meaningful experience. Definitions of MW have tangentially considered [116] the temporal structure of experience, although it has been a common denominator in experimental subjects’ narratives. An experimental and conceptual issue for future research is to assess the relevancy of temporal experience and determine how the constitutional processes of subjectivity and intersubjectivity relate to the dynamic interaction of the large-scale brain networks, i.e., in terms of the patterns of activity and ‘silence’ of the triple network and other large scale networks, relative to the unfolding patterns of the constitutive processes. For future research, the latter could be a point to develop as an example of the reciprocally enriching benefits between the biomedical sciences and the humanities and social sciences.

The functional connectivity research seems to lack a conceptual framework to account for subjectivity and self and a model or grounded hypothesis which defines them. Based on the twofold process, a framework for a model of subjectivity and the ‘self’ could guide the design of accurate and relevant protocols to address better the challenges posed by psychopathological manifestations. To face the latter challenges, understanding the constitutional function of temporal and movement experience is necessary. Complementarily, the latter framework would also require the integration of the function of the cerebellum, specifically regarding the ventral and dorsal projections to the cortex through the dentate nucleus, inferior olive nucleus ([117], p. 6) and the basal ganglia caudate nucleus [118].

To conclude, this paper contributes to the psychopathological tradition that identifies temporal experience as a grounding factor of mental disturbances by hypothesizing that a disruption of the twofold constitutive function of kinesthesia (indication) and touch/proprioception (presentation) contributes to the pathogenesis of schizophrenia. These ideas have to be enhanced conceptually, developed experimentally, and tested clinically, and their feasibility has to be assessed and adjusted to those professional domains through collaboration with functional connectivity research, brain imaging technology, and pharmacological clinical research.

## Data Availability

Not applicable.

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
