# Peer review of "Kinesthesia and Temporal Experience: On the ‘Knitting and Unknitting’ Process of Bodily Subjectivity in Schizophrenia"

_diagnostics, 2022, doi:10.3390/diagnostics12112720_

Round 1

Reviewer 1 Report

The manuscript is interesting but rather long. If possible, some parts could be synthetized in order to improve readability. So, before resubmitting ask yourself if all parts are strictly necessary to sustain the thesis that phenomenological insight are useful to improve neuroscientific research in the field.

Minor changes required:

a) p. lines 232-233. The concepts of passive synthesis and conation are not obvious. The latter is defined below, at p.7 l.277 and f. Probably they should both defined at their first mention.

b) p. l.235. The world “Into” was part of a sentence that was deleted?

c) p.10 l.401. Instead of “Psychiatrists defined …” write “Minkowski writes”, because those are his own concepts of schizophrenia, not those of psychiatrists in general.

Author Response

Many thanks for your useful comments. The whole paper has been synthesized and edited both regarding length and readability, it is over 2000 words shorter now. Footnotes have been also shortened. In addition, the readability was improved by introducing more sections with clearly defined scopes and headings. We have also wrote a completely new abstract giving a better account of the paper as a whole. Last but not least, the paper was edited regarding English grammar. 

More specifically, we have now defined the concepts of passive synthesis and conation. The whole on Minkowski was also edited and condensed, and your specific suggestions were implemented. 

Many thanks again for reviewing our paper, and we hope that its current version will be satisfactory. 

Reviewer 2 Report

This is a narrative review article that tries to articulate a conceptual analysis of the Kinaesthesia concept and with some relevant data in terms of its neurobiological correlates. In our view, the abstract should be reformulated in order to briefly summarize the contents of the article (and not just mention what the article would be about). On the other hand, some qualitative ways of referring to authors as "guided by the great philosopher Edmund Husserl" (page 1,); "Minkowski, a polish jew mostly educated in ...." should be removed as this is a scientific article and not a textbook.
In the discussion section, it would also be important to clearly summarize the main conclusions of this article, which, being quite dense, needs a summary and clear conclusion.

Author Response

Many thanks for your useful comments that helped us to improve the paper.

-The abstract was written anew to give a better and clear account of the paper as a whole. 

-The references to historical authors have been edited and synthesized 

-The discussion section was clarified and extended. 

Overall, the paper was synthesized and shortened by over 2000 words for better readability. Sections have also been introduced with clear headlines to help navigate the narrative. Many paragraphs and footnotes have been shortened or synthesized. Finally, the whole paper was checked for English grammar and vocabulary to improve readability.